# Health in Chile’s Recent Constitutional Process: A Qualitative Thematic Analysis of Civil Proposals

**DOI:** 10.3390/ijerph192416903

**Published:** 2022-12-16

**Authors:** Baltica Cabieses, Sophie Esnouf, Alice Blukacz, Manuel A. Espinoza, Edward Mezones-Holguin, René Leyva

**Affiliations:** 1Programa de Estudios Sociales en Salud, ICIM, Facultad de Medicina Clínica Alemana, Universidad del Desarrollo, Santiago 7610315, Chile; 2Departamento de Salud Pública, Escuela de Medicina, Pontificia Universidad Católica de Chile, Santiago 8331150, Chile; 3Centro de Excelencia en Investigaciones Económicas y Sociales en Salud, Universidad San Ignacio de Loyola, Lima 15024, Peru; 4Instituto Nacional de Salud Pública de México, Cuernavaca 62100, Mexico

**Keywords:** public health, Chile, democracy, civil participation, reform, qualitative research, Latin America

## Abstract

(1) Background: In response to the recent political crisis in Chile, the “Agreement for Social Peace and the New Constitution’’ was approved. We aimed to analyze the health-related civil proposals uploaded to the official website for popular participation in the new constitution in Chile. (2) Methods: We carried out a qualitative thematic analysis of 126 health-related valid proposals. Moreover, we analyzed their link to the Health Goals 2030, established by the Ministry of Health of Chile and to the Sustainable Development Goals (SDGs). (3) Results: Sixteen main categories were reached. In all, they were organized into four main areas: (i) the right to health and the establishment of a universal health system; (ii) effective access to selected healthcare services; (iii) improving health outcomes for all and for the relevant subgroups; and (iv) the social determinants of health, health in all the policies, and community health. We found that these four areas were strongly linked to the Health Goals 2030 for Chile and to the SDGs. (4) Conclusions: Despite the fact that the new constitutional proposal was rejected in September 2022, the civil health-related proposals and the areas of health and healthcare were of interest to the citizens as the request showed a strong demand from the population for participation in matters of health, healthcare, and public health.

## 1. Introduction

### 1.1. Chile: An Unequal Country with a Fragmented Health System

Chile is a high-income country, with a GNI per capita of USD 24,020 in 2020 [1]; however, 20.7% of its population, or over 3.5 million people, experiences multidimensional poverty [2], and the proportion of people experiencing income poverty grew between 2017 and 2020 from 8.6% to 10.8% of the population after a sustained drop over the preceding decade [3]. With regard to healthcare, the health system is fragmented between the public system, called the National Health Fund (the Spanish acronym is FONASA), which covers 76.5% of the population, and the private health care system (the Spanish acronym is ISAPRE), covering 15.4%. The armed forces and police system is separate from both of these systems and covers a little over 2% of the population. Additionally, the percentage of the population reporting not being covered increased from 2.8% in 2017 to 4.3% in 2020 [4].

In terms of the provision of care, the institutions covered by FONASA are public healthcare centers managed by the National System of Healthcare Services (the Spanish acronym is SNSS). The institutions covered by the ISAPREs are privately owned healthcare centers or independent healthcare providers, and the armed forces and police system has dedicated centers [5]. Additionally, there is a growing market for complementary private health insurance. In contrast to the public health insurance system, the private health insurance companies can reject applicants if their financial contribution does not match their estimated health risk or simply because they suffered diseases in the past and are at a high risk of new events in the future [6]. In this context, in many cases, effective access to healthcare may depend on the ability to pay for private care, due to inefficiencies in the underfunded and ill-equipped public system, which explains a high proportion of out-of-pocket expenditure in Chile [7,8,9]. From a social determinant of health perspective, experiencing multidimensional poverty and income poverty leads to poorer health outcomes throughout the course of life, especially in a country where the healthcare system is highly unequal and fragmented between two main systems which are vastly different with regard to accessibility and quality of care [10]. The existing inequities in this health system have even led to international ethical discussions in this regard [11].

The Chilean healthcare system went through a major reform in 2005 during President Ricardo Lagos’ term of office (2000–2006). The reform aimed at guaranteeing equal access to healthcare for high priority health problems to people residing in Chile according to their needs and without discrimination [12]. The healthcare reform was implemented in 2003 and defined a set of health interventions that, according to the System of Health Guarantees Law, should be provided to every person that required them in Chile, irrespective of the type of provision entitlement, the ability to pay, or any other non-need factor. The policymakers who designed and implemented this reform expected it to produce a significant impact on the population’s health [13]. However, social inequalities concerning access to healthcare and the population’s health grew significantly over time, especially for those whose health conditions were not prioritized in the AUGE/GES law [14,15].

Regardless of the changes introduced in the 2005 reform, Chile continued to have a socioeconomically segmented healthcare system, which deepened the inequities derived from risk selection and cream-skimming, which are the typical problems of relatively unregulated insurance markets. Thus, the structure of the system ended up with two different subsystems, one for the relatively more socioeconomically advantaged citizens and another for the relatively more disadvantaged, poorer people. This organization contributed to generating a perception of injustice based on the fact that richer people had timely access to many health services, whereas poorer people had to deal with waiting lists, more restricted access to technologies, and a perception of worse health services [16]. Moreover, differences in effective access to healthcare continued to be reported in the country, based on the type of healthcare insurance, gender, socioeconomic status, borough and region of residence, and other relevant social determinants.

### 1.2. Social Unrest and Chile’s Constitutional Process

A sustained period of social, economic, and political unrest took place during October 2019 in Chile, known as the “Social Outburst” or Estallido Social (in Spanish), where, for weeks, mobilization and strikes took place. The outburst was triggered by a rise of CLP 30 for public transport (around USD 0,034, a third of a dollar), and people’s demands revolved mainly around the deep inequalities concerning pensions, health, and education in Chile [17,18,19]. In response to the serious political crisis, which previously led to public disorder and violence, Chile’s political parties came together to deliberate on an institutional solution and attend to the social demands. The result was the elaboration of the “Agreement for Social Peace and the New Constitution’’ (Acuerdo Por la Paz Social y la Nueva Constitución) [20], a document in which the signing parties subscribed to “guarantee their commitment with reestablishing peace and public order in Chile, with total respect to human rights and the current democratic institutions (…) through an unobjectionably democratic procedure”. It was signed on the 15th of November 2019, by most of the political parties. The parties agreed on the initiation of a constitutional process that, if successful, would culminate in a new constitution that would replace the existing one, written in 1980 during Augusto Pinochet’s dictatorship period.

Thereafter, on the 25th of October 2020, an initial national plebiscite was held, where voters decided on: (i) whether they approved or rejected the drafting of a new constitution and (ii) what body should oversee writing it. More than half of eligible voters participated, and the results were: (i) 78% of voters approved the creation of a new constitution and 21% rejected it, and (ii) 79% voted for a democratically elected constitutional convention and 21% voted for a mixed constitutional convention. Previously, on the 11th of April 2020, elections to choose the members of the constitutional convention were held [21].

### 1.3. Citizen Participation in the New Constitution

Once the body was elected and formed, a term of 9 to 12 months was established to work on the draft for a new constitution. During this process, to promote citizen participation, the Popular Initiative for Norms (from Iniciativa Popular de Norma in Spanish) platform was created. This initiative acted as a participation mechanism through which an individual or a group of people could present proposals for constitutional norms to the constitutional convention. A total of 6114 proposals were sent through an online platform and underwent an admissibility revision, after which 2496 were approved and published for the public to vote on; each person could support a total of seven proposals. Physical centers were also enabled so that those with no internet access could inscribe their proposals manually. To be eligible to participate, the people were required to be above 16 years of age, of Chilean nationality, foreign nationality with Chilean residence, or Chileans living abroad. They then had to register in a Single Register of Popular Participation (*Registro Único de Participación Popular*), which could be performed online or at one of the enabled centers [22].

Of these proposals, 10.7% were submitted by social organizations or private institutions and 89.3% were sent by individuals. Following the two-week voting period, 77 proposals (3%) reached the required 15,000 votes or more to pass on to the next stage of discussion. Of these, 60 were supported by organizations or institutions and 17 by individuals [23].

### 1.4. Popular Participation in Health Policy and Health-Related Civil Proposals for the New Constitution

Citizen participation in public health initiatives has historically been in the form of social movements. These share a common cause and advocate for change in matters such as urban conditions and health, the health of children, and the behavioral and substance-related determinants of health, amongst others [24]. Health social movements (HSM’s) have been described those which “challenge state, institutional and cultural authorities in order to enhance public participation in social policy and regulation”, and they can play an important role in influencing health policy [25]. The particularity of the participation mechanism enabled during the constitutional process was that not only could these movements put forth their ideas, but so could individuals who did not form part of a collective and whose perspectives had not been perceived before.

As described before, inequality in health was one of the main social demands brought up by citizens during the period of social unrest in Chile. In this context, a survey carried out by the Center for Conflict and Social Cohesion Studies (COES), in December 2019, asked people to assign a score from 1 to 10 regarding the importance of the different social demands, with 10 being very important. A total of 89.5% of people assigned 10 points to health [26,27,28]. Similarly, a survey carried out by the Center of Public Studies (CEP) during that same period showed that people placed “bad quality public health and education” in the fourth position when asked what they thought to be the main reason for social manifestations in Chile [29].

Additionally, the last National Health Survey (2021) [30] showed that the attributes which were considered to be the most important in building a dignified care system were: having a public and private system of equal quality; having equal access to health for everyone, independently of the socioeconomic situation; that everyone could feel economically safe if they got sick; and having timely access to health, eliminating the waiting time for important situations. Based on these results, we assumed that most proposals would be associated with the addressing of these inequalities.

On this basis, we conducted a narrative revision and synthesis of all the available civil proposals put forth as constitutional norms concerning health in order to better understand the most relevant areas of demand and interest for Chileans concerning their health care and their suggestions on how it could be improved. Our general research objective was to describe the main public health themes represented in this process in the country. By describing these themes and identifying the priorities put forth by the civil population, this study can be useful in guiding future reforms, both constitutional reforms and those regarding the health system. The association of these proposals with wider frameworks allows to visualize whether the institutional priorities are aligned with the popular demands and, if so, how. Despite the fact that the new proposal for the constitution was finally rejected during Chile´s referendum in September 2022, this historical societal, democratic, and participatory experience deserves regional and global attention and analysis.

## 2. Materials and Methods

Study design: We carried out a qualitative thematic analysis of the health-related civil proposals uploaded to the official website for popular participation in the new constitution in Chile.

Data source: The data used were extracted from the “Digital Platform for Popular Participation”, an online, public website created to enable citizen participation during the Chilean constitutional process. The approved proposals were published in the section called “Popular Initiative for Norms”. Each proposal was structured following these headings: Problem at hand; Ideal situation; What the new constitution must contemplate; Article proposal; and Brief background on the person, group, or institution in charge of the initiative.

Search strategy: The keywords ‘health’, ‘healthy’, ‘well-being’, ‘quality of life’, and ‘health systems’ were used in the website’s search engine, which displayed the proposals containing these words in any part of the text. The data search and the data retrieval were conducted by SE and BC. The data search and the selection were conducted between January and April 2022.

Selection criteria: A total of 145 proposals were founded. A first revision was made based on the title of each proposal, excluding those duplicated and those not relevant to the subject; following this, 126 remained. Previously, a complete reading of these proposals was made, and those whose main content related to health or public health were selected according to their relevance; after which, 106 proposals remained and were fully analyzed for extraction.

Data extraction: The extraction of data was conducted by entering the data into a Microsoft Excel table by SE; the data were classified by (i) the general characteristics of the proposal (proposal number, amount of votes, author/name, supporting institution, three keywords, central ideal of the proposal, audience it would benefit); (ii) the relevance to public health (does it mention public health in the title? yes/no; does it mention public health within the text? yes/no); (iii) the mention of Essential Public Health Functions (EPHF), as established by the WHO/PAHO, 2020; and (iv) the mention of additional elements: “interculturality”, “gender”, “right”, or “vulnerability” or, similarly, “global”, “social security”, “health system”, “social”, or “civil participation”. The data extraction was conducted by SE, and the data analysis was conducted by SS and BC.

Data analysis: We conducted narrative qualitative analysis [31], in which researchers collect descriptions of events or happenings and then configure them in meaningful ways. The content related to each of the mentioned pre-defined categories was carefully investigated and extracted into the Excel sheet Verbatim. After this was finalized, the emerging codes were identified within each category of analysis, which allowed for these unique codes to be described in detail. After this descriptive stage, we oversaw the findings to identify the content that was potentially duplicated across categories. This was very rare, and when it happened, we located the content in the pre-defined category that was most significantly represented. In this way, we avoided repetition of findings and allowed for a general organization of the qualitative data.

Data synthesis and display: Based on these results, sixteen main categories were reached using an inductive approach. These categories were: Mental health; Dental health; Public employees in health; Child and adolescent health; Autonomous health and education; Health, sustainability, and environment; Women, sexual and reproductive health rights; Palliative, chronic pain, and end-of-life care; Medication, specific treatments, and alternative therapies; Occupational medicine; Rare diseases and catastrophic situations; Promotion and prevention in health; Waiting times in health; Citizen participation; Health coverage and financing; and Right to health and the universal health system. In all, they are organized into four main areas: (i) the right to health and the establishment of a universal health system; (ii) effective access to selected healthcare services; (iii) improving health outcomes for all and for relevant subgroups; and (iv) social determinants of health, health in all the policies, and community health.

Additionally, each proposal is linked to the relevant Health Goals 2030 established by the Ministry of Health of Chile and the relevant Sustainable Development Goals (SDGs) established by the 2030 Agenda, and this detailed information is presented together with the full list of proposals in Appendix A. The following section will describe each of them.

Figure 1 illustrates the alignment of the proposals with the Health Goals 2030 and ten of the SDGs. Table 1 connects all 16 categories with the 4 overarching identified dimensions.

## 3. Results

The sixteen categories reached were grouped into four main areas which organize the results.

### 3.1. The Right to Health and the Establishment of a Universal Health System

#### 3.1.1. Right to Health and Universal Health System

The thirteen proposals grouped into this category suggest that the new constitution must establish health as a right, with different approaches as to how this will be achieved. “Single, Universal, Plurinational and Integrated Health System for the New Chile that we are building democratically” had 19,852 votes, making it the most popular in this category; it passed on to the next phase of discussion and was supported by the Cabildo “Health, a right”. It puts forth that the new constitution must allow for the transition to a single, universal, plurinational, and integrated health system that replaces the fragmented and inequitable system, where health access is based on subsidiarity, economic capacity, and the profit of the providers.

“Health for Chile that includes everyone” had 17,294 votes; it passed on to the next phase of discussion and was supported by the National Confederation of Municipal Health Workers (CONFUSAM). It suggests that health be recognized as a universal human right and that there should be a unique and universal health system, based on solidarity and participation, which incorporates views on gender, interculturality, and decentralization. The proposal states that this system should be financed by progressive and proportional taxes and that it should be based on primary care attention. Similarly, “FENPRUSS and the right to health” obtained 16,460 votes; passed on to the next phase of discussion and was supported by The National Confederation of University Professionals of Health Services (FENPRUSS). In addition to establishing a right to health, this proposal advocates that health systems must focus on the social determinants of health and must be based upon the principles of equity, universality, solidarity, comprehensiveness, interculturality, quality, efficiency, and gender. It puts forth that there should be a universal health system financed by a single, national fund.

“Universal Health System” (4598 votes) puts forward that the state must guarantee access to decent and equal health for all people, and for this, there must be a universal health system. Along the same lines as these proposals were “For decent health as a fundamental right for Chile” (842 votes); “Right to health: First world medicine for everyone” (751 votes); “Right to health: autonomy, dignified death, universal health system, binding participation, dignified attention and right to care” (572 votes); “A national health system for Chile: territorial, public, of quality and free” (424 votes); “Universal health system from the moment of birth for the duration of life for all citizens, independently of their socioeconomic situation” (416 votes); “Right to public and private health” (340 votes); “Health is a right” (195 votes); “Universal Health, latest generation hospitals, how and what way to achieve and implement it in Chile” (192 votes); and “’El Bosque’ borough demands a universal health system conceived as a human right for everyone” (191 votes). They all concur that health should be established as a fundamental right and that the state plays an important role in ensuring access to proper healthcare conditions.

The proposals are all linked to Health Goal 7. Management, quality, and innovation and the proposal “Health for a Chile that includes everyone”; they also include Health Goal 1. Environmental health and healthy environments. With regard to the SDGs, they are all connected to SDG 3. Good Health and Wellbeing; SDG 10. Reduced Inequalities; and SDG 16. Peace, Justice, and Strong Institutions, except for “Health is a right”, which is not explicitly connected to SDG 16. Additionally, “FENPRUSS and the right to health” is also linked to SDG 5. Gender Equality.

#### 3.1.2. Health Coverage and Financing

Currently, in Chile, there are two main options for health insurance. On one hand, there is the National Health Fund (FONASA), which acts as a public agency that collects and manages the compulsory contributions of employees. Alternatively, people may opt for private insurance through the private healthcare insurers ISAPREs, which is often more expensive. The present constitution states that people have the right to choose between these two. “The right to choose your insurance, health and education in public and private systems” (358 votes) and “Pension system and health insurance” (128 votes) aim at maintaining this principle.

Other proposals suggest reforming the current system, and they propose establishing a single, public, and universal plan, where private insurances are a complementary option. This is the case for “Your health, your payment, your right” (3410 votes); “Health without patches: Free, transparent and dignified” (1935 votes); “Right to Health” (1842 votes); and “Public, integrated health system” (382 votes). On the other hand, “Strengthening the health system, so that all Chilean people have access to quality and timely health benefits in an equitable manner” (990 votes) puts forth that the state must ensure universal health coverage, although it does not define how this should be carried out. Similarly, “To eliminate the obligatory health percentage of people in ISAPRES, leaving FONASA as the sole receptor” (220 votes) and “You FONASA, me FONASA, we are all FONASA” (162 votes) specify that there must be a compulsory contribution from every citizen to FONASA, independently of whether they have private insurance.

Other financial proposals include “Right to health and the presence of profit” (96 votes), which suggests eliminating all profit from both public and private health systems, and “0.1 of taxes to social security and health” (25 votes), which puts forward that 0.1% of the taxes collected by the state must be directed to a common fund that goes to health (to improve and maintain medical centers, emergency services, etc.) and to another fund that ensures minimum guaranteed pensions.

All the proposals are connected to Health Goal 7. Management, quality, and innovation; regarding the SDGs, they are connected to SDG 3. Good Health and Wellbeing; SDG 10. Reduced Inequalities; and SDG 16. Peace, Justice, and Strong Institutions. The proposal “0.1 of taxes to social security and health” is also the only one to be connected to Health Goal 6. Emergencies and disasters.

#### 3.1.3. Waiting Times in Health

Two proposals were grouped into this category. “Diminish waiting times in access to public health services” (404 votes) says the state must be expedient in helping citizens access health attention and medication, and in cases where there is no availability for a certain service or medication, they must allow access to other professionals or systems. “Guarantees for prolonged waiting times in health” (173 votes) refers to the state’s responsibility in providing aid in cases where waiting for health attention is prolonged. Both proposals are connected to Health Goal 7. Management, quality, and innovation and to SDG 3. Good Health and Wellbeing and SDG 10. Reduced Inequalities.

### 3.2. Effective Access to Selected Healthcare Services

#### 3.2.1. Mental Health

This category held the highest number of proposals, with thirteen proposals alluding exclusively to mental health and two combining mental and dental health. The most popular, “Right to free and quality mental health” (20,590 votes), puts forward that “citizens will have the right to free and optimal mental health services”, and ten other proposals support this notion.

“Right to mental health with no discrimination, in equal conditions to physical health” (1175 votes) and “Free, quality mental health as a fundamental right, and decentralized priority for Chileans” (323 votes) explicitly mention free mental healthcare. “A right to mental health, promotion of wellbeing, dignified and inclusive mental health services. May no one is left behind” (1755 votes); “Mental health as a right and not a privilege” (1699 votes); “Mental health as a constitutional norm: Guaranteeing the universal right to protection of mental health” (644 votes); “Inalienable right to prevention, detection, diagnosis, attention, treatment and education in mental health” (298 votes); and “The state’s responsibility of encouraging, promoting and guaranteeing quality mental health attention for all Chileans” (293 votes) mention the state’s responsibility in guaranteeing the fulfilment of mental health as a right. “Timely access to free, quality mental health. A patient should never have to say ‘I have been waiting for an appointment for months’” (827 votes) and “Right to quality mental health for everyone” (537 votes) both refer to equal, timely and quality access to mental health.

Both “Free physical, mental and dental health” (418 votes) and “Guaranteed access to quality and free mental and dental health at a community level, for everyone, independently of their socioeconomic level” (169 votes) integrate mental and dental health. They state that the new constitution must include free and quality access to these services. On the other hand, “Right to community mental health: closure of psychiatric hospitals” (378 votes) and “Towards the integration of communitarian mental health” (70 votes) specifically touch on community-based mental health, highlighting the importance of cultural relevance to the territory and shifting towards a communitarian, as opposed to individualistic, approach to health. In terms of integrating mental health into other spaces, “Strengthening mental health” (774 votes) argues that public policy must be aimed at the “promotion of mental health in education, health and work areas, as well as equal treatment conditions”.

With regard to alignment with wider frameworks, the proposals related to mental health mainly relate to the following Health Goals 2030: 1. Environmental health and healthy environments; 2. Lifestyle; 4. Non-communicable chronic diseases and violence; 5. Development and disability; and 7. Management, quality, and innovation. Additionally, all the proposals feed into SDG 3. Good Health and Wellbeing and SDG 10. Reduced Inequalities. In addition, the proposal titled “Right to mental health with no discrimination, in equal conditions to physical health” also contributes explicitly to SDG 5. Gender Equality.

#### 3.2.2. Dental Health

In addition to the two proposals that integrate dental and mental health, six proposals exclusively focus on dental health, with “Right to dental health” (2964 votes) being the most popular initiative. “Oral health as a right” (665 votes); “The right to oral health for all Chileans” (529 votes); and “Oral health as a fundamental right for the elderly population” (809 votes) all allude to the same concept, with the last one, in particular, specifying the importance of including the elderly population. On the other hand “Dental health, urgently” (438 votes), though not alluding to the concept of rights, aims to guarantee dental health to anyone who needs it, in a similar way to “Dental health guaranteed in case of loss of dental pieces” (169 votes).

The dental health proposals are aligned with the following Health Goals 2030: 4. Non-communicable chronic diseases and violence and 7. Management, quality, and innovation, insofar as they address oral health, which is included in Goal 4. This is an argument in favor of including the right to dental health as a part of healthcare, signifying changes in the management of the system. The proposal “The right to oral health for all Chileans” also pushes forward a community approach to the topic, which relates to Health Goal 1. Environmental health and healthy environments. Concerning the SDGs, all the proposals are aligned with SDG 3. Good Health and Wellbeing and SDG 10. Reduced Inequalities.

#### 3.2.3. Occupational Medicine

The three proposals included in this category refer to the consecration of health and safety in the workplace as a fundamental right and they are all related to Health Goals 1. Environmental health and healthy environments; 2. Lifestyle; and 7. Management, quality, and innovation, as well as SDG 3. Good Health and Wellbeing; SDG 8. Decent Work and Economic Growth; and SDG 10. Reduced Inequalities. These are: “Including health and safety at work (SST) as a fundamental right for workers” (1353 votes); “Safety and health at work is a fundamental right according to OIT (International Labour Organisation). It is necessary to include it in the constitution” (244 votes); and “Right to safety and health at work” (189 votes).

#### 3.2.4. Palliative, Chronic Pain, and End-of-Life Care

The four proposals grouped into this category mainly concern the delivery of end-of-life care, as well as the working and training conditions of healthcare workers. “Care and health protection in the new constitution” (1332 votes), supported by the National Federation of Nurses of Chile, suggests incorporating the right to receive care by making it a state duty to ensure its provision, while the “Right to ‘good death’” (496 votes) and “Mental health and wellbeing of elderly patients and their careers. Professionalizing the care of the elderly” (269 votes) refers particularly to the ensuring of the adequate training of health teams and professionals dedicated to caring. “Right to live without pain”: for compassionate treatment of chronic, non-oncological pain” (226 votes) specifically refers to the right to live without chronic pain and argues that the constitution must ensure the effective right of people to have integral physical and mental health.

In terms of alignment with Health Goals, these proposals all relate to at least Health Goals 1. Environmental health and healthy environments and 7. Management, quality, and innovation. However, “Care and health protection in the new constitution” is also linked to 2. Lifestyle and 4. Non-communicable chronic diseases and violence, while the other three have to do with 5. Development and disability. All four proposals are linked to SDG 3. Good Health and Wellbeing and SDG 10. Reduced Inequalities.

#### 3.2.5. Medication, Specific Treatments, and Alternative Therapies

Fifteen proposals were grouped into this category, which includes the proposals related to access to medication and treatment and different forms of therapies.

Medication:

This is associated with medication and pharmacies; “Access to quality medication and pharmacies as health centres” (6026 votes) and “Medication as an essential article of public use, and pharmacies as primary health centres” (330 votes) suggest that access to medications must be guaranteed and consider medications to be essential goods. Likewise, they both suggest that pharmacies should acquire a more central role in the healthcare system by providing primary care. “Universal and free access to all remedies and treatments, independently of the disease” (1210 votes), as its name states, looks to establish and guarantee a system of universal and free access to all remedies and medications, independently of the cost of the disease, financed by the state.

Specific treatments:

Regarding specific treatments, there were six heterogeneous proposals. The most popular proposal was “Obesity and loose skin: our health is your responsibility” with 12,936 votes, and it raised the argument that the state should be responsible for ensuring corrective surgery prior to abdominoplasties, along with promoting healthy lifestyle habits. On a similar note, there was a proposal named “Right to quality esthetic health for everyone”; however, it only obtained 92 votes. With regard to access to physical therapy, “Right to physical therapy and rehabilitation to face disability-School of Physiotherapists” (8225 votes), supported by the Chilean School of Physiotherapists, puts forward that the state must guarantee timely and adequate access to good-quality physical therapy and rehabilitation for all the citizens in the national territory. Two proposals referred to organ donation, “Organ donation and transplants and their promotion as a public policy” (1234 votes) and “Creation of a Nation Organ Transplant Institution” (154 votes). The first makes the state responsible for ensuring organ transplants for those who need them, whereas the second proposes the creation of a separate institution that coordinates organ transplants, guaranteeing transparency and autonomy. Lastly, “Blood as a common good and its equal and opportune availability for all inhabitants of the country” (706 votes) aims to create policies that will ensure timely, efficient, and equal blood transfusion in the country.

Alternative therapies:

Seven proposals referred specifically to alternative or complementary forms of therapy. “Cannabis to the Constitution now: For the right to freedom in the development of personality, personal sovereignty and wellbeing” counted 44,332 votes. It was the proposal with the highest number of votes concerning health, and it was third in the overall category. It favors the free choice to use cannabis and other psychoactive substances of natural or synthetic origins, in alignment with a balanced and respectful bond with nature. Six other proposals refer to the inclusion of alternative forms of treatment into health systems, supporting the idea that people must be guaranteed a right to choose and to access these forms of medicine should they choose to. These include: “Acupuncture-Chinese Medicine-Integrative therapies” (4140 votes); “Floral therapy as a constitutional right” (1603 votes); “Health and natural medicine: a necessity for Chile” (1493 votes); “Health centred on the human being, with integrative and complementary medicine” (802 votes); “Right to access health care through conventional, traditional or complementary medicine, according to choice” (387 votes); and “Single, a diverse and universal health system with recognition of non-conventional medicine and plurinational” (118 votes).

In the medication subcategory, the proposals are all connected to Health Goals 1. Environmental health and healthy environments; 3. Communicable diseases; 4. Non-communicable chronic diseases and violence; and 7. Management, quality, and innovation. In the specific treatments’ subcategory, all the proposals are linked to at least Health Goals 4. Non-communicable chronic diseases and violence and 7. Management, quality, and innovation, and some include 1. Environmental health and healthy environments and/or 5. Development and disability. Finally, the subcategory alternative/complementary therapies are all connected to Health Goals 1. Environmental health and healthy environments; 2. Lifestyle; and 7. Management, quality, and innovation.

With regard to the SDGs, while all the proposals are aligned with SDG 3. Good Health and Wellbeing and SDG 10. Reduced Inequalities, the proposal “Cannabis to the Constitution now: For the right to freedom in the development of personality, personal sovereignty and wellbeing” also aligns with SDG 11. Sustainable Cities and Communities, SDG 13. Climate Action, and SDG 15. Life on Land.

#### 3.2.6. Rare Diseases and Catastrophic Situations

The three proposals grouped into this category establish that the coverage of costs associated with rare diseases or catastrophic/terminal situations must be guaranteed. “Mandatory and guaranteed coverage of health insurance in cases and diagnosis of rare diseases” (2866 votes) makes the insurance (public or private) responsible for covering these costs in treatments associated with rare diseases. “Right to health by way of treatment of high-cost diseases” (413 votes) and “Elective and free health for the catastrophically ill” (115 votes) refer to the same concept but in the context of catastrophic and terminal conditions. All three proposals are connected to Health Goals 5. Development and disability and 7. Management, quality, and innovation, as well as SDG 3. Good Health and Wellbeing and SDG 10. Reduced Inequalities.

### 3.3. Improving Health Outcomes for All and for Relevant Subgroups

#### 3.3.1. Child and Adolescent Health Proposals

Five proposals were grouped into this category. They all specifically refer to children and adolescents as a particular subgroup within the population. “Guaranteed mental health for our children and youth” (513 votes) and “Every child has a right to have a good quality of life and the state must propitiate and guarantee adoption systems and capacitation centres” (137 votes) address vulnerability and suggest reinforcing adoption systems and children’s mental health to provide safe environments and prevent future delinquency. “Equalitarian and independent access to health care for children and adolescents” (53 votes) refers to facilitating the youth’s access to healthcare systems and advocates for removing the requirements of obtaining a parent or tutors’ consent to access health services. Lastly, “Healthcare within educational institutions” (830 votes) and “Strength conditioning and functional training for middle school students, to strengthen physical and mental health and gender equality” (224 votes) bring up the role of schools and educational institutions and suggest health promotion and prevention strategies that could be carried out from within those spaces.

All the proposals on child and adolescent health are aligned with at least Health Goal 1. Environmental health and healthy environments and 7. Management, quality, and innovation, as well as SDG 3. Good Health and Wellbeing and SDG 10. Reduced Inequalities. Some proposals also relate to Health Goals 2. Lifestyle; 4. Non-communicable chronic diseases and violence; and/or 5. Development and disability, as well as SDG 4. Quality Education and/or SDG 5. Gender Equality.

#### 3.3.2. Women, Sexual and Reproductive Health Rights

Eight proposals were grouped into this category. In general, the proposals point towards the importance of sexual and reproductive rights, including access to adequate healthcare and contraceptive methods, as well as the promotion of education in this area. “It will be a law” (38,198 votes) together with “Right to sexual and reproductive health with a gender, feminist, intersectional and pluralist focus” (15,558 votes) favor abortion, and both were approved for the next phase of discussion. The first proposal defends the idea of abortion being included as a constitutional right; the second proposal suggests a new legal system that guarantees sexual and reproductive health. In contrast, “Always for life” (31,208 votes) puts forward that the constitution must protect life from the moment of conception and that the law will protect the life of unborn beings. On a similar note, “Conscientious objection to abortion for healthcare workers”, which aims to give healthcare workers the option to not be part of abortion-related procedures for ethical or personal reasons, received 351 votes.

Four other proposals are in line with sexual and reproductive rights: “Guarantee the right to voluntary contraception within the framework of protecting sexual and reproductive rights” (615 votes) promotes the notion of free and quality access to contraceptive methods; “¡Right to sexual health!” (712 votes), which additionally incorporates the importance of training students on sexual health as well as providing sexual education for the general population; “Access to permanent menstrual education and health for girls, women and menstruating people in Chile” (217 votes), which specifically refers to menstrual health and education; and lastly, “Reproductive rights and access to assisted reproduction in the new Chile: a proposal from civilians and scientific societies” (4492 votes), supported by the Chilean Society of Reproductive Rights, puts forward assisted reproduction, suggesting it should be accessible to all people and couples. Finally, “Respected gestation and birth” (375 votes) proposes that doulas should be incorporated into the current health system, as a support for women during different stages of their gestation, birth, and postpartum cycles.

The proposals all relate to Health Goals 2. Lifestyle, which explicitly includes reproductive and sexual health, and 7. Management, quality, and innovation, as well as SDG 3. Good Health and Wellbeing, SDG 5. Gender Equality, and SDG 10. Reduced Inequalities.

#### 3.3.3. Public Employees Proposals

Four proposals were grouped into this category: “Public authorities and staff must use public health, education and transport services” (2911 votes); “State health care for all public workers” (1074 votes); “Government authorities and staff must use public health and education systems” (611 votes); and “Quality public health” (295 votes). They all suggest the compulsory use of public services, including health services, by government authorities and public employees, arguing that this will generate awareness of how they work and will therefore act as an incentive for them to be improved. Not dissimilarly, “Equal health for all” (225 votes) puts forward that all health professionals must provide care to patients covered by the public insurer FONASA, without exclusion.

All the proposals are aligned with Health Goals 1. Environmental health and healthy environments and 7. Management, quality, and innovation, as well as SDG 3. Good Health and Wellbeing and SDG 10. Reduced Inequalities.

### 3.4. Social Determinants of Health, Health in All Policies, and Community Health

#### 3.4.1. Autonomous Organisms for Health and Education

All three proposals in this group refer to the establishment of autonomous entities to manage health and educational matters. “Autonomous health and education” (91 votes); “Autonomy for education and health (85 votes)”; and “Establishment of autonomous organisms for health and education (31 votes)” all concur that having institutions that function independently of the government and the changing political authorities will allow for long-term projects and goals to be reached, and finally, they are all aligned with Health Goals 1. Environmental health and healthy environments and 7. management, quality, and innovation, as well as SDG 3. Good Health and Wellbeing, SDG 4. Quality Education, and SDG 10. Reduced Inequalities.

#### 3.4.2. Health, Sustainability, and Environment

Five proposals were grouped into this category. “Health and quality of life” (6005 votes) and “Right to a good living through the protection against acoustic contamination” (370 votes) focus on acoustic contamination, its impact on physical and psychological health, and the importance of proper regulation to ensure a better quality of life. “The right to health in an environment free of contamination” (712 votes) and “The right to health in an environment with reduced contamination”—both supported by the United for Responsible Technology institution—address the adverse effects of contaminants present in different technologies, and the need for this to be regulated by the state to ensure people’s wellbeing. Finally, “Without a right to food, there is no right to health nor healthy development” (108 votes) refers to the importance of healthy and sustainable food, both in the effort to implement healthier habits and in the effort to be environmentally friendly.

All the proposals clearly relate to Health Goal 1. Environmental health and healthy environments, and the proposal “Without a right to food, there is no right to health nor healthy development” can also be linked to 2. Lifestyle and 4. Non-communicable chronic diseases and violence. Regarding the SDGs, all feed into SDG 3. Good health and Wellbeing and SDG 10. Reduced Inequalities; however, additional SDGs become relevant in specific proposals: SDG 2. Zero Hunger, SDG 10. Reduced Inequalities, SDG 11. Sustainable Cities and Communities, SDG 13. Climate Action, and SDG 15. Life on Land.

#### 3.4.3. Promotion and Prevention in Health

The six proposals grouped into this category aim at incorporating health promotion and prevention as fundamental pillars of health and wellbeing. Regarding health promotion, “Better health, lifestyle and wellness through physical education, exercise and recreation (187 votes)” and “Incentivized Health” (45 votes) both highlight the importance of incentivizing sports and physical activity. The first proposal is more elaborate than the other one and suggests that this could be achieved by increasing the number of hours allocated to physical education in schools, guaranteeing the promotion of physical and recreational activities for families, encouraging the active use of free time and leisure, and enabling adequate environments for physical activity and exercise for the general population.

In terms of preventive measures, there are three proposals. The first one, titled “Constitution and Health: Modernization of the State” (289 votes), suggests that prevention should be at the center of public health policies and that the state should establish a “National Preventive Health System”. On a similar note, the proposal “Right to health access (Disease prevention or free checkups)” (270 votes) suggests that the state must guarantee access to health services, both public and private, with a preventive focus, and “Preventive health as a fundamental right” (162 votes) argues that preventive health must be guaranteed as a fundamental right and a constitutional principle, focusing on nutrition and the elimination of sedentarism. Lastly, “Right to knowledge and formation in first aid and basic health concepts” (130 votes), looks to guarantee education on first aid skills and aims at the prevention of high-incidence diseases in Chile.

With regards to Health Goals, all proposals are aligned with at least 1. Environmental health and healthy environments and 2. Lifestyle, and some are also related to 3. Communicable diseases, 4. Non-communicable chronic diseases and violence, 5. Development and disability, and/or 7. Management, quality, and innovation.

#### 3.4.4. Citizen Participation in Health

Four proposals specifically refer to the participation of citizens and communities in decision making in health. “Right to an integral and participative health: creating wellbeing from the communities” (541 votes); “For a right to health based on the community” (483 votes); and “Binding citizen participation in health” (184 votes) put forward that communities must be systematically involved in decision making and that there must be sustained participation in areas concerning health. “Towards an open, binding and feminist health” (87 votes), suggests a series of reforms and changes, both ideological and structural, that incorporate a transversal perspective on health and include active, binding citizen participation.

The proposals are connected to Health Goal 7. Management, quality, and innovation and SDG 3. Good Health and Wellbeing and SDG 10. Reduced Inequalities. The last proposal, “Towards an open, binding and feminist health” also relates explicitly to SDG 5. Gender Equality.

## 4. Discussion

Our study reviewed the proposals linked to health pushed forward by civil society for their inclusion in the new constitution of Chile through the online platform “Iniciativa popular de norma” (Popular Initiative for Norms). After careful consideration, 126 proposals were selected for review and classified into 16 categories. This study analyzed these proposals at a historical time for Chile, in which civil participation in the elaboration of the new constitution was promoted. We conducted this narrative synthesis of all the available civil proposals put forth as constitutional norms concerning health to better understand the most relevant areas of demand and interest for Chileans concerning their health care and their suggestions on how it could be improved.

The first important finding, in line with our research assumptions based on the last National Health Survey, has to do with the proposals linked to the right to health and the establishment of a universal health system. Health was indeed one of the main issues at hand during the 2019 “estallido social”, and one of the grievances expressed was the fragmentation of the healthcare system between the public and the private sector, where the private healthcare sector operates for high profit and on a discriminatory basis according to the patient’s capacity to match contribution with estimated needs, while the public healthcare is understaffed and underfunded and struggling to respond to the healthcare needs of the majority of Chile’s population [7,32,33]. That the right to health and the establishment of a universal healthcare system are highly represented among the civil proposals is then not surprising, and while the proposals are diverse, they all point towards the state as having the main responsibility for providing, or at the very least guaranteeing, adequate access to acceptable care. This demand is also expressed in some of the 11 proposals which focused on health coverage and financing, several of which advocate for a unique and universal funding system for healthcare. However, some proposals argue in favor of maintaining the segmented system framed as the right to choose between one or the other sector. Regardless of this minority of pro-status quo proposals, it can be argued that most proposals put forward deep, structural changes about how health is funded, provided, and conceptualized at the constitutional level. Furthermore, some proposals related to health also extend to other, more specific issues, such as the waiting times to receive care, medication, palliative care, and dental healthcare, which are also among the grievances expressed during the “estallido social”.

The second main finding of this review, in a similar line, is the number of proposals related to mental health, which are as many as that of the one more broadly focused on the right to health. Mental health is a key issue in Chile, considering, on the one hand, that the prevalence of suicide and common mental disorders in Chile is high [34,35] and, on the other hand, that these issues remain largely unaddressed by the healthcare system due to structural gaps, lack of funding, and prioritization [32,36,37]. Introducing the explicit right to mental health in the constitution rather than keeping it implicit as simply part of the right to health would turn it into a priority at the policy level; however, improving the delivery of mental healthcare and creating the conditions to guarantee health environments that foster mental health will require deep, structural changes involving more than simply the health sector. In that sense, the proposals put forward reforms to the healthcare system to guarantee health as a right, with guidance on how to implement them, which is paramount for achieving many of the other changes proposed, such as guaranteeing the right to mental health at a constitutional level.

The third main finding is that beyond only healthcare, many proposals also include the need to address the social determinants of health and to approach health from a community perspective. The Chilean public health system is characterized by a family and communitarian approach to primary care, which strives to be patient-centered and delivers integral and continuous care [38], something that is often jeopardized by the realities of an overworked and underfunded system, and by the co-existence of a private, for-profit healthcare system which is very much at odds with these principles. Again, achieving the changes pushed forward by the proposals is arguably dependent on other structural changes. Additionally, it is interesting that alternative and complementary medicine are represented in several proposals and linked to principles of cross-cultural care, as well as the development of a respectful relationship with nature. The dominant medical system in Chile is the biomedical system, despite being a diverse country with around 2 million people reporting claiming indigenous ethnicity, among whom are Mapuche, Aymara, Quechua, and Rapa Nui, with their medical cosmovision and systems [39]. Although significant efforts to promote cross-cultural dialogue in health have been made, through, for instance, the Indigenous Health Policy in the early 2000s, this has not signified integral recognition of their medical system [40]. This lack of recognition was implicit in the Popular Initiative for Norms, in which there was no option given to submit proposals in languages other than Spanish; yet, it is important to note that the constitutional convention had 17 out of the 155 seats reserved for indigenous members, a first in terms of the political representation of indigenous people in political decisions in Chile.

Now, as much as the Popular Initiative for Norms appeared to be a far reaching and inclusive participation mechanism, the fact is that how these proposals would then be managed by the constitutional convention raised much criticism. The proposals were to be discussed by members of the convention and then put to the vote amongst them, leaving no relevance to the number of votes from the public or to other aspects and leaving the final call up to the political conformation of the convention, where left and center-left representatives predominated. This raises questions on how effective this democratic mechanism is if it incises the final draft of the new constitution and how representative of the nation’s interest it is.

In association with this last point, although we found that the proposals associated with health were multiple, diverse, and with a general trend towards improving quality and access in health nationwide, various themes that are still deficient in our health system were missed. One example of this is migrant health. Not one proposal made specific mention to this group and the challenges they face in our national health system. Administrative and cultural barriers continue to perpetuate their vulnerability in Chile [41]; yet, this was not raised during the constitutional process. This notion forces us to attentively examine which groups, causes, or initiatives were not included or given light to and to consider how we can work towards giving a voice to those who still do not have the tools or opportunities to do so.

Civil proposals represent an important effort from civil society, either grassroots organizations, institutions, or individuals, towards having a say in the formulation of the new constitution. However local this effort might be, it can also be placed in a greater policy framework at the national level—the Health Goals 2030 established by the Ministry of Health of Chile—and in another framework at the global level, the SDGs. Regarding the Health Goals 2030, it is worth noting that most of the proposals are linked to Goal 7. Management, quality, and innovation, suggesting that changes in the way in which care is guaranteed and delivered are especially needed. Furthermore, virtually all the proposals are connected to SDG 3. Good Health and Wellbeing and SDG 10. Reduced Inequalities. These aspects of discussion that emerged from this unique study of civil proposals in Chile is informative on how society perceives and makes demands for the selected public health strategies in the context of a highly unequal high-income country in the Latin American region. This is the case for many countries worldwide as they share the complex condition of simultaneously dealing with the health issues of sub-populations that experience chronic conditions and cancer as well as those of other sub-groups that continue to suffer the health effects of poverty and exclusion, including historical and re-emerging infectious diseases. The segmented and fragmented health system in Chile is struggling to deal with multiple and competing health needs, and the people of the country are demanding more universal coverage and more equity-based planning and effective delivery at the point of care. The existing design of the health systems in almost all countries in Latin America is being challenged, and in some cases, the idea of a health system reform is currently under discussion, as it is in the case of Chile.

These proposals represent a call for a definite departure from the current constitution, in which the constitutional rights to health are only very sparingly defined as the right to choose between public and private coverage. Although it is unlikely that all aspects brought up in the many proposals reviewed will make it into the final constitution, considering the current inequities in both physical and mental health in Chile, the new constitution must be a catalyst for change in how health is approached and guaranteed. Furthermore, this process contributes to tearing down the authoritarian legacy of the Pinochet dictatorship. It is a process that was a long time coming and establishes another crucial precedent after the return to democracy in 1990, where people have the opportunity to update one of the most important frameworks guiding the country. More specifically, with regard to health and healthcare, the proposals show a strong demand from the population to have a say in matters of health, healthcare, and public health.

This study has a number of limitations and strengths. In terms of its limitations, it describes only the civil proposals that were effectively uploaded in the national platform. Hence, it represents successfully submitted proposals, not all the attempts to submit proposals in the process, and the effective number of incomplete submissions is impossible to know as there is no recollection of them in the system. Moreover, individuals or organizations could only upload a maximum of seven proposals; so, there might be additional relevant proposals that were not prioritized and, therefore, are not represented in this analysis. Despite these limitations, this study is the first of its kind to analyze the proposals related to health that were uploaded and voted for during the elaboration of the new constitution for Chile. This national process is remarkable in its spirit of promoting wide civil participation. From our thematic analysis, we were able to describe the main areas of interest and the issues of concern of the citizens in the country, most of which were related to the right to health and the establishment of a universal health system, improving mental health, and the need to address the social determinants of health and to approach health from a community perspective. These three main areas of relevance to this analysis are strongly linked to broader global public health concerns, including the Health Goals 2030 and the SDGs.

## 5. Conclusions

Chile is a high-income country in Latin America that faces great socioeconomic inequality and a segmented–public and private–health system. In response to the serious political crisis, Chile’s political parties came together recently to deliberate on an institutional solution, attending to the social demands of fairness around several issues, such as pensions, health, and education. The result was the elaboration of the “Agreement for Social Peace and the New Constitution’’. In this context, we conducted a narrative revision and synthesis of all the available civil proposals put forth as constitutional norms concerning health to better understand the most relevant areas of demand and interest for Chileans concerning their health care and their suggestions on how it could be improved. We believed that, despite the fact that the proposal for the new constitution was finally rejected during Chile´s referendum last September 2022, this historical societal, democratic, and participatory experience could be relevant in providing information on the societal interests related to public health matters in highly unequal high-income countries in the Latin American region, with Chile as a case study.

Our general research objective was to describe the main public health themes represented in this process in the country. We found 16 main categories: Mental health; Dental health; Public employees in health; Child and adolescent health; Autonomous health and education; Health, sustainability, and environment; Women, sexual and reproductive health rights; Palliative, chronic pain, and end-of-life care; Medication, specific treatments and alternative therapies; Occupational medicine; Rare diseases and catastrophic situations; Promotion and prevention in health; Waiting times in health; Citizen participation; Health coverage and financing; Right to health and a universal health system. In conclusion, the main areas of interest and issues of concern of the citizens in the country are related to four relevant dimensions: the right to health and the establishment of a universal health system; effective access to selected healthcare services; improving health outcomes for all and for relevant subgroups; and the social determinants of health, health in all the policies, and community health. These four main areas were also strongly linked to broader global public health concerns including the Health Goals 2030 and the SDGs.

## Figures and Tables

**Figure 1 ijerph-19-16903-f001:**
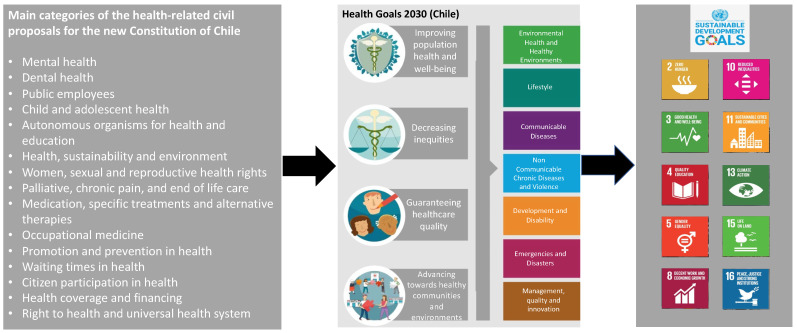
Health-related civil proposals for the new constitution, Health Goals 2030, and SDGs.

**Table 1 ijerph-19-16903-t001:** Classification of the popular initiatives by dimensions and categories of analysis. Health-related civil proposals for the new constitution in Chile, 2022.

General Dimension	Main Category
i. The right to health and the establishment of a universal health system	1 Right to health and universal health system
	2 Health coverage and financing
	3 Waiting times in health
ii. Effective access to selected healthcare services	4 Mental health
	5 Dental health
	6 Occupational medicine
	7 Palliative, chronic pain, and end-of-life care
	8 Medication, specific treatments, and alternative therapies
	9 Rare diseases and catastrophic situations
iii. Improving health outcomes for all and for relevant subgroups	10 Child and adolescent health
	11 Women, sexual and reproductive health rights
	12 Public employees in health
iv. Social determinants of health, health in all the policies, and community health	13 Autonomous health and education
	14 Health, sustainability, and environment
	15 Promotion and prevention in health
	16 Citizen participation

## Data Availability

Not applicable.

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
