# Peer review of "Health in Chile’s Recent Constitutional Process: A Qualitative Thematic Analysis of Civil Proposals"

_ijerph, 2022, doi:10.3390/ijerph192416903_

Round 1

Reviewer 1 Report

Cabieses and colleagues conducted a qualitative, thematic analysis of 126 health-related proposals and their link to the Chile's Health Goals 2030 and the 20 Sustainable Development Goals (SDGs). The study identified 16 main categories that were re-organised in four main areas, which were strongly linked to broader global public health concerns. 

The authors may wish to add more details on who conducted the literature search, data retrieval, and thematic analysis, etc., and how the "inductive approach" was actually performed (lines 183-184). 

Please consider shortening the length of the Conclusions section (lines 766-804). 

Overall, it seems to be a bit difficult for international readers to interpret the study findings if one is not very familiar with the health-care context of Chile. Beside, most of the literature cited in the current paper were closely related to Chile alone. Alternatively, the authors may wish to discuss the relationship of their main study findings and take-home messages with that reported in other similar studies conducted elsewhere. This may enhance the generalisability of the current study findings to other countries or health-care settings which face comparable challenges in fragmented care and underfunded public system. 

Reviewer 2 Report

The article addresses an important and interesting issue, especially in the Chilean context, where privatisation of health and education has been a constant problem and a "good example" for Latin America to imitate; without thinking, this problem has deepened the socio-economic inequalities in the region. So, congratulations, and thank you for analysing this problem that has done so much damage to the population's health.

Introduction

The Introduction clearly states the research problem and the importance of the study. However, some essential elements are missing, such as: 

-       State of the art on the topic (what is known about popular health policy initiatives?)

-       The contribution of the study (what the study and results are for?)

-       The research questions and objectives (this appears only in the Abstract)

-       The research assumptions or tentative ideas of the study (this is the equivalent of the research hypothesis in quantitative studies)

It is helpful to end the Introduction with these elements.

Methods

I suggest revising the name of Table 1: a) Dimensions and categories for organising popular initiatives, b) Classification of the popular initiatives by dimensions and categories…

The following methodological elements are also unclear:

-       Which author supports the qualitative thematic analysis of your study?

-       When were the data collected? This is equivalent to fieldwork

-       What was the data analysis strategy used? ¿Is it discourse analysis, content analysis, or narrative analysis? ¿Is qualitative thematic analysis the same as narrative analysis? Which author supports this similarity or difference?

Results and Discussion

The first sentence/paragraph is already stated several times in the text, and I suggest saying here that the dimensions and categories of the study organise the results. 

The authors mention the popular initiatives and the number of supporters they have received. In addition, the authors relate the popular initiatives to the Health Goals 2030. However, what is the results of the study? Is it the registration of the popular initiatives and the number of endorsements received by each? This is not a result of the study because this existed on the website where the authors collected the data. Are the results merely recording how the Chilean people´s popular health initiatives align with the Health Goals 2030?

In this sense, the results should be presented with an emphasis on what have been the main health concerns of the Chilean people in the debate over the new constitution. It is necessary to better show in the results what is included and excluded in the discussion because classifying or organising the initiatives and support received in four dimensions and 16 categories is not a result. 

A few lines should be devoted to analysing the role of democracy in improving public health systems: what factors have caused some issues to be included in the debate and not others? Have the initiatives been really popular? What part of the population supports them, and what part rejects them? Are the contents of the initiatives linked to the contents of the media? Or are they related to the agenda of some political party? If a narrative study is made, it is mandatory to look at the meaning of these contents.

It is also interesting to know the socio-demographic profile (age, gender, class, place of residence, ethnic, religious, political-ideological, sexual identity, etc.) of the people supporting the initiatives and to compare them with each other. If you have access to these data, it is essential to incorporate them into the analysis because they are the elements that define health inequalities.

One more question to problematise the issue: is it only possible to carry out the initiatives through the digital platform? If so, has this been a problem for the participation of some population groups? For example, older people tend to have less digital communication skills, has this affected participation?

On the other hand, have all the initiatives been done in Spanish, or is it possible to do them in native languages? The authors consider it not essential because the speakers of native languages (such as Mapuche, Quechua or Aymara) are bilingual and speak Spanish? There are just ideas to reflect on the linguistic situation of the native peoples of the region. 

¿Why do you think some health problems are not included in popular initiatives? What do other authors say about the Chilean case or other similar contexts? This is essential to the Discussion. 

Despite this suggestion, the Discussion is well-structured and sufficiently broad. However, these questions could be considered because the Discussion is a space for theoretical debate and dialogue with other authors and contexts. 

Conclusions

You should limit yourself to answering the research question, stating if the study achieved the proposed objectives and how it was done, and not repeating the dimensions and categories that organise the results.

One last question: what are the main limitations of the study? All studies have limitations.  
